# Determination of Trace Thorium and Uranium Impurities in Scandium with High Matrix by ICP-OES

**DOI:** 10.3390/ma16083023

**Published:** 2023-04-11

**Authors:** Zhixuan She, Minglai Li, Zongyu Feng, Yang Xu, Meng Wang, Xijuan Pan, Zhiqing Yang

**Affiliations:** 1National Engineering Research Center for Rare Earth, GRINM Group Co., Ltd., Beijing 100088, China; 2GRIREM Advanced Materials Co., Ltd., Beijing 100088, China; 3General Research Institute for Nonferrous Metals, Beijing 100088, China; 4GRIREM HI-TECH Co., Ltd., Sanhe 065201, China; 5Hebei Province Rare Earth Functional Materials Manufacturing Innovation Center, Xiong’an, Baoding 071700, China

**Keywords:** scandium oxide, high purity, ICP-OES, Th, U

## Abstract

High-purity scandium oxide is the principal raw material of high-purity scandium metal and aluminum scandium alloy targets for electronic materials. The performance of electronic materials will be significantly impacted by the presence of trace amounts of radionuclides due to the increase in free electrons. However, about 10 ppm of Th and 0.5–20 ppm of U are typically present in commercially available high-purity scandium oxide, which it is highly necessary to remove. It is currently challenging to detect trace impurities in high-purity scandium oxide, and the detection range of trace thorium and uranium is relatively high. Therefore, it is crucial to develop a technique that can accurately detect trace Th and U in high concentrations of scandium solution in the research on high-purity scandium oxide quality detection and the removal of trace impurities. This paper adopted some advantageous initiatives to develop a method for the inductively coupled plasma optical emission spectrometry (ICP-OES) determination of Th and U in high-concentration scandium solutions, such as spectral line selection, matrix influence analysis, and spiked recovery. The reliability of the method was verified. The relative standard deviations (RSD) of Th is less than 0.4%, and the RSD of U is less than 3%, indicating that this method has good stability and high precision. This method can be used for the accurate determination of trace Th and U in high Sc matrix samples, which provides an effective technical support for the preparation of high purity scandium oxide, and supports the production of high-purity scandium oxide.

## 1. Introduction

Scandium (Sc) is not a lanthanide element, but it is classified as one of the rare earth elements (REEs) in a broad sense because of its similarities to lanthanide elements in terms of outer structure and chemical properties, and scandium is also considered one of the dispersed elements. Due to its widespread application in high-tech products, scandium is referred to as a high-tech metal element. Scandium-related products are employed in the fields of electronic communications, laser materials, and electric light source materials [1,2,3,4,5,6]. Therefore, scandium is regarded as a strategic resource by the United States, the European Union, Japan, etc. The largest consumption of scandium products in China is Al-Sc alloy. The high-purity Al-Sc alloy target is the key material in high-end fields such as Radio Frequency (RF) filters and chips [7]. The performance of the Al-Sc alloy target’s application is significantly influenced by its purity. The material’s performance will be significantly affected by the presence of some trace impurities, and it may even have an impact on how devices and equipment function normally. When the high-purity Al-Sc alloy target is created using the physical vapor deposition (PVD) process for integrated circuits, for instance, the decay of trace impurity of the radionuclides thorium (Th) and uranium (U) continuously releases high-energy particles while the chip is in operation. These particles are easily ionized by electrons, exceed the chip’s breakdown voltage, and result in the chip’s fatal failure.

High-purity scandium oxide is the principal raw material for preparing high-purity scandium metal and aluminum–scandium alloy targets. Commercial high-purity scandium oxide frequently contains impurities Th 10 ppm and U 0.5–20 ppm, making it challenging to meet the demand for high-purity scandium oxide from high-tech products. The fundamental issue to be solved is how to efficiently extract Th and U from high-purity scandium oxide. The primary methods used nowadays to separate and purify rare earth oxides are solvent extraction, ion exchange, extraction resin chromatography, etc. Among them, the ion exchange method provides the most benefits for producing high-purity rare earth [8,9,10,11]. It can also be used for the separation of rare earth elements such as scandium and some difficult-to-separate elements with analogous properties, such as the separation of impurities such as Th and U from high-purity scandium. It is crucial to develop an expeditious, accurate, and reliable test method for detecting Th and U impurities throughout the separation, purification, and application of high-purity scandium oxide.

At present, the analytical methods for U and Th mainly include the volumetric method, spectrophotometry, plasma atomic emission spectrometry, etc. [12,13,14,15,16,17]. The advantages of inductively coupled plasma atomic emission spectrometry (ICP-OES), such as its low detection limit, high sensitivity, high precision, small interference, simple and quick testing, and simultaneous determination of multiple elements, have led to its rapid development and widespread application. ICP-OES is also the most convenient method among the ones just mentioned, owing to its capacity to estimate multiple elements simultaneously in a pretty short time. Additionally, it has the lowest limit of detection and the highest accuracy of detection, compared with the traditional methods. Therefore, ICP-OES is also widely used in the determination of thorium, uranium and other actinides. ICP-OES, for identifying various elements in scandium oxide, such as Th, is introduced in GB/T 13219-2018 [18], for instance. The determination range of Th oxide content is 0.0005~0.10%, but this method cannot determine the content of uranium oxide in scandium oxide; GB/T 18114.2-2010 introduces the method for determining the content of thorium oxide in rare earth concentrate by ICP-OES [19]. The range for determining the thorium oxide content is 0.30–8.00%, which is not appropriate for determining trace thorium oxide, and the method cannot determine the content of uranium oxide in scandium oxide. The simultaneous determination of Th and U content in Sc, the primary component element, by ICP-OES has not yet been reported. As the primary component of high-purity scandium oxide in this instance, the content of Sc is significantly higher than that of the impurities Th and U. The high concentration of Sc will seriously interfere with the identification of Th and U when ICP-OES is utilized for testing. Therefore, the establishment of a rapid detection method for trace Th and U in the background of high Sc has imperative practical significance for the separation and purification of trace impurities Th and U in high-purity scandium oxide and the quality detection of high-purity scandium oxide.

In this study, inductively coupled plasma emission spectrometry was chosen to simultaneously detect radionuclides Th and U in the high-purity scandium oxide system that contains trace Th and U, in the absence of internal standard elements. Methods such as analytical spectral line selection, matrix impact analysis, and spiked recovery test were used for accurate determination.

## 2. Experimental

### 2.1. Instrument

To determine the Th and U in the samples, an Agilent 5800 ICP-OES was utilized (Agilent Technologies Co., Ltd., Beijing, China). Glassware, pipette, and other rudimentary instruments for analysis and testing were also utilized. Sample propulsion was accomplished using a three-channel peristaltic pump and PVC peristaltic pump pipe. The operating conditions of the ICP-OES system are summarized in Table 1.

### 2.2. Reagents

Stock standard solutions (1000 mg∙L^−1^) of Th and U were used (Guobiao Testing & Certification Co., Ltd., Beijing, China).

Preparation of Sc matrix stock solution (50 g∙L^−1^): 7.667 g of scandium oxide powder was weighed [relative purity ω((Sc_2_O_3_)/(∑REO)) ≥ 99.9995%, ω(ΣREO) ≥ 99.5%]. It was then transferred into a 200 mL beaker, along with 30 mL of concentrated hydrochloric acid, heated at 200 °C without bumping until completely dissolved. It was then cooled to room temperature, transferred to a 100 mL volumetric flask, diluted with water to the scale, and thoroughly mixed; 1 mL of this solution contains 50 mg Sc. The hydrochloric acid was super pure, and the water for experimental analysis was ultra-pure water prepared by an ultra-pure water mechanism (resistivity 18.25 MΩ∙cm).

### 2.3. Standard Series Solution

Generally, the higher the concentration of matrix in standard solution, the lower the limit of determination, under the condition that Th and U occupy a certain proportion in the matrix. The sample does not have to be diluted too much to make the concentration of thorium and uranium too low. On the other hand, the plasma emission spectrometer’s injection system (nebulizer, rectangular tube, etc.) can quickly become blocked by the injection analysis of a continuous high-concentration solution, which reduces the atomization effect and causes a significant variation in the test results. It is not desirable to set the Sc matrix concentration too high in order to guarantee the steady detection of multiple samples. In the standard series of solutions configured in this work, 5 g∙L^−1^ Sc was used as the matrix solution, and the concentration range of Th and U was 0–0.0125 g∙L^−1^, which not only ensured the stability of the test but also reached the lower limit of determination needed for the study. The standard series of solutions are shown in Table 2.

### 2.4. Experimental Solution

High-purity scandium oxide weighing 0.5000 g was placed in a 100 mL beaker with hydrochloric acid, heated until completely dissolved, allowed to cool to ambient temperature, and was then transferred to a 100 mL volumetric flask with a specific quantity of Th and U. They were mixed thoroughly after diluting with water to the scale. The experimental solution with Sc concentration of 5 g∙L^−1^ and Th, U concentration of 0.025, 0.125, 0.25, 1.25, 2.5 and 12.5 mg∙L^−1^, respectively, was prepared.

## 3. Results and Discussion

### 3.1. Selection of Analytical Spectral Lines

Two to four spectral lines with higher sensitivity were chosen for each element in accordance with the analytical lines provided by the literature and Chinese national standards [18,19,20], such as 367.007 nm and 409.013 nm for U, and 269.242 nm, 283.232 nm, 283.730 nm, 339.203 nm and 401.913 nm for Th.

Figure 1 depicts the scanning spectrum after scanning different solution systems with various wavelengths (5 g∙L^−1^ Sc; 2.5 mg∙L^−1^ Th; 2.5 mg∙L^−1^ U; 5 g∙L^−1^ Sc and 2.5 mg∙L^−1^ Th, 2.5 mg∙L^−1^ U). The scanning spectrum provides a preliminary understanding of the effects of the Sc matrix and U on the Th scanning spectrum. As shown in Figure 1, 5 g∙L^−1^ Sc matrix and U have a certain impact on the detection of Th at the wavelength of 269.242 nm; at the wavelength of 283.232 nm, the interference of Sc matrix and U on Th measurement is small, at the wavelength of 283.730 nm, U interferes with the measurement of Th to some extent. However, the intensity of 2.5 mg∙L^−1^ of U is less than 3700 a. u., which is much lower than the 32,500 a. u. high intensity of 2.5 mg∙L^−1^, leading to the limited influence of U. At a wavelength of 339.203 nm, Sc and U do not interfere much with the test, but the intensity of Th is too low to be detrimental to the test. At a wavelength of 401.913 nm, however, the Sc matrix and U have a significant impact on the measurement of Th, and the intensity of Th is insufficient.

Different solution systems (5 g∙L^−1^ Sc; 2.5 mg∙L^−1^ Th; 2.5 mg∙L^−1^ U; 5 g∙L^−1^ Sc and 2.5 mg∙L^−1^ Th, 2.5 mg∙L^−1^ U) were scanned at different wavelengths, and the scanning spectrum is shown in Figure 2. From the scanning spectrum, a preliminary understanding of the impact of the Sc matrix and Th on the U scan spectrum is possible. Figure 2 shows that the spectral line of U at a wavelength of 367.007 nm is greatly affected by Sc, and the spectral line of Sc has completely covered the spectral line of U. However, in the diagram of the spectral line of U scanned at the wavelength of 409.013 nm, the Sc matrix and Th had little interference on the measurement of U, and the spectral intensity of U is high.

The standard series solutions were scanned by ICP-OES in turn, and the spectral lines with higher linearity of standard curve, smaller relative standard error (RSE), more symmetrical peak shape, and less spectral interference were selected as analysis lines. Figure 3 illustrates that the spectral peak form of Th at 283.730 nm is more symmetrical and consistent, with superior linearity and a relative standard deviation of 7.31%, than that at 269.242 nm, 283.232 nm, 339.203 nm, and 401.913 nm. The spectrum at 269.242 nm, in contrast, shows a heterogeneous peak, and the Th peak shape is asymmetric and of low intensity. At wavelengths of 283.232 nm, the spectra have hybrid peaks with lower intensities than at wavelengths of 283.730 nm. The spectra have a miscellaneous peak at 339.203 nm, and the Th peak has a low intensity, an asymmetrical shape, and a high relative standard error, which prevented them from falling within the range of relative standard error needed for linear fitting. At 401.913 nm, the spectra have a miscellaneous peak, and the Th peak shape is non-symmetrical and low intensity.

Figure 4 shows how significantly the Sc matrix affects the scanning line of U at a wavelength of 367.007 nm to which is attributed the intensity of Sc at more than 400,000 a. u. It is unable to conduct linear fitting because the intensity of U is too low and the spectral line is obscured. At 409.013 nm, the spectral peak shape of U is symmetrical and linear, and the RSE is 18.75%.

Following a review of the calibration curve and spectrum described above, the spectral lines with high linearity, symmetrical peak shape, high sensitivity, low spectral interference, low background and high signal-to-background the ratio were selected as analytical lines, and the wavelengths of ICP-OES Th and U in this work were determined to be Th 283.730 nm and U 409.013 nm, respectively.

### 3.2. Precision Test

According to the established test method, the prepared test solution (as described in Section 2.4) was repeated 10 times, and the standard deviations (SD) and relative standard deviations (RSD) of Th and U were calculated based on the test results, respectively, as shown in Table 3. The findings of the precision test indicate the relative standard deviation RSD of Th < 0.4% and the relative standard deviation RSD of U < 3% in each sample. It is evident that the test procedure and ICP-OES instrument have excellent precision and stability.

### 3.3. Trueness of Test Method

The spiked recovery of Th and U were evaluated on five groups of samples to assess the test method’s trueness. The results of this investigation into the test error and the spiked recovery rate are provided in Table 4. It can be seen from Table 4 that the test procedure has good trueness and that the recovery rate of the spiked elements Th and U can be adjusted between 90.00% and 107.00%.

### 3.4. Limit of Detection and Limit of Quantification

Ten parallel tests were carried out on the zero standard blank sample (containing 5 g∙L^−1^ Sc matrix, and the concentration of Th and U was 0 g∙L^−1^), and the standard deviations (SD) of Th and U were calculated. The limit of detection and the limit of quantification of this element by this approach are, respectively, three times the standard deviation (3 SD) and ten times the standard deviation (10 SD), respectively. Table 5 displays the results. As can be observed, the detection limit of this approach is less than 0.000015 g·L^−1^ for Th and 0.000060 g·L^−1^ for U. The quantitation limits for Th and U are less than 0.00005 g·L^−1^ and 0.00020 g·L^−1^, respectively.

### 3.5. Influence of Sc Concentration on Th and U Detection

It is essential to look into how the Sc matrix concentration affects the outcomes of Th and U detection because of the high Sc content in the test solution of the high-purity scandium oxide impurity elimination research technique. In order to create test solutions with the same concentrations of Th and U, the stock solution of the Sc matrix and the standard solution of Th and U were utilized. The concentrations of Sc were then set at 0, 1, 2, 3, 4, 5, and 6 g∙L^−1^, respectively (as shown in Table 6). This group of solutions was tested using the above method, and the test results are shown in Table 7. The test’s results indicate that the measured Th and U content in the samples without the Sc matrix (Sample No. 1) is higher than the actual levels. The relative error is at a pretty high level, up to 30.8% for Th especially. This is due to the existence of characteristic lines caused by the 5 g/L scandium matrix in the spectra of standard series solutions. If there is no scandium matrix in the sample or the concentration of the scandium matrix is different from that in the standard series solutions, it would not be able to match the spectral lines of the standard series solutions, resulting in the difference between the detected value and the true value. We can find from the data in the Table 7 that, the closer the scandium ion concentration in the sample is to the scandium matrix concentration in the standard series solution, the smaller the difference between the measured value and the true value of thorium and uranium content, and the smaller the relative error would be. As the concentration of the Sc matrix in the test liquid increases, the test results for the Th and U contents gradually decline, and when the concentration of the Sc matrix increases to 6 g∙L^−1^, the test values of Th and U contents are lower than the true values. The measured contents of Th and U are the most accurate when the concentration of the Sc matrix is 5 g∙L^−1^, or when the concentrations of the Sc matrix in the test solution and standard solution are equal, indicating that the concentration of the Sc matrix has a great influence on the measurement of Th and U. It is therefore required to dilute or concentrate the solution to be tested to the Sc concentration of 5 g∙L^−1^ before the procedure can be utilized for accurate testing in the detection of the quality of high-purity scandium oxide and the study of the removal of radioactive Th and U.

## 4. Conclusions

The primary components Sc, Th, and U are present at very different concentrations in high-purity scandium oxide during the separation and purification of the trace impurities Th and U, and the matrix effect has a significant impact on the Th and U testing accuracy. In this paper, a method for the simultaneous and accurate determination of Th and U impurities by ICP-OES was established.

(1)According to this study, the detection limits for each element in the established analysis and test method are a Th less than 0.000015 g∙L^−1^ and a U less than 0.000060 g∙L^−1^, respectively. The recovery rates for each element are between 90.00% and 107.00%, and the relative standard deviation RSD for Th and U are both less than 0.4% and 3.0%, respectively.(2)Before using this method for an accurate test, it is required to dilute or concentrate the test solution to be consistent with the concentration of the Sc ion in the standard solution (Sc concentration 5 g∙L^−1^), as the concentration of the Sc matrix has a significant interference on the test of Th and U.

In high matrix Sc, this approach simultaneously detects trace Th and U impurities. The analysis and test guarantee for the development of new technology for the separation and purification of trace Th and U radionuclides in high-purity scandium oxide, as well as the quality detection and evaluation of high-purity scandium oxide, are made possible by the test’s accuracy, quickness, and simplicity, which significantly reduces test workload, increases research efficiency and provides analysis and test guarantees.

## Figures and Tables

**Figure 1 materials-16-03023-f001:**
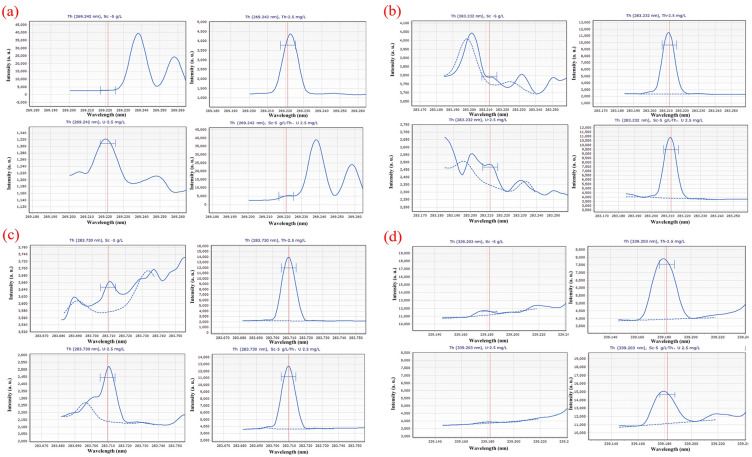
Scanning spectra in different solution systems (5 g∙L^−1^ Sc; 2.5 mg∙L^−1^ Th; 2.5 mg∙L^−1^ U; 5 g∙L^−1^ Sc and 2.5 mg∙L^−1^ Th, 2.5 mg∙L^−1^ U) at (**a**) 269.242 nm; (**b**) 283.232 nm; (**c**) 283.730 nm; (**d**) 339.203 nm; (**e**) 401.913 nm wavelength. Where the red vertical straight lines represent the wavenumbers of the peaks; the blue dashed lines represent background signal curves; the blue curves represent the signal curve of the sample; and the symbols “H” are target peak position identifiers. The same below.

**Figure 2 materials-16-03023-f002:**
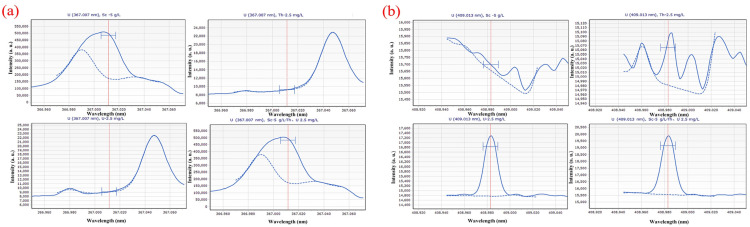
Scanning spectra in different solution systems (5 g∙L^−1^ Sc; 2.5 mg∙L^−1^ Th; 2.5 mg∙L^−1^ U; 5 g∙L^−1^ Sc and 2.5 mg∙L^−1^ Th, 2.5 mg∙L^−1^ U) at (**a**) 367.007 nm; (**b**) 409.013 nm wavelength.

**Figure 3 materials-16-03023-f003:**
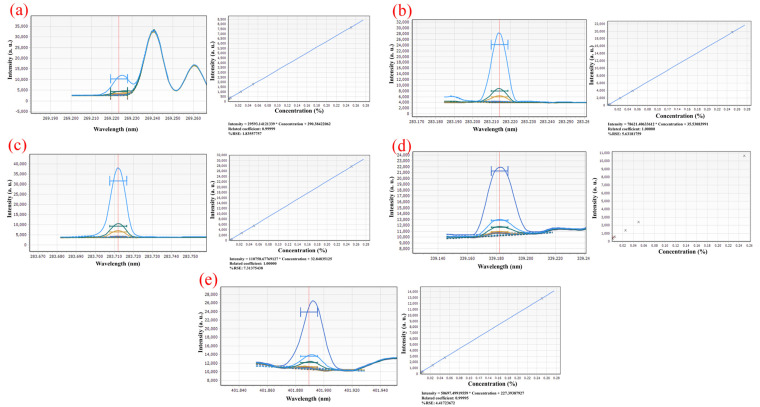
Element spectrum of Th in (**a**) 269.242 nm; (**b**) 283.232 nm; (**c**) 283.730 nm; (**d**) 339.203 nm; (**e**) 401.913 nm.

**Figure 4 materials-16-03023-f004:**
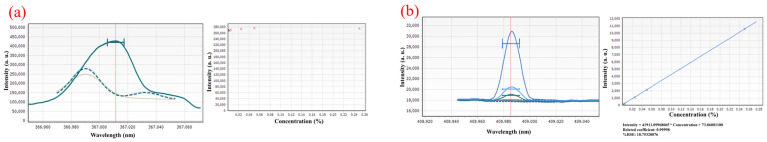
Element spectrum of U in (**a**) 409.013 nm; (**b**) 367.007 nm.

**Table 1 materials-16-03023-t001:** Operating conditions for ICP-OES.

Parameters	Value
Pump speed	20 rpm
Read time	5 s
Plasma flow rate	12 L/min
Auxiliary gas flow rate	1 L/min
Boost latency	20 s
Radiofrequency incident power	1.2 kW
Stabilization time	20 s
Viewing mode	radial

**Table 2 materials-16-03023-t002:** Standard series solution and concentrations.

**Element**	**Standard** **Sample #0**	**Standard** **Sample #1**	**Standard** **Sample #2**	**Standard** **Sample #3**
Sc	5 g∙L^−1^	5 g∙L^−1^	5 g∙L^−1^	5 g∙L^−1^
Th	0 mg∙L^−1^	0.000025 mg∙L^−1^	0.000125 mg∙L^−1^	0.00025 mg∙L^−1^
U	0 mg∙L^−1^	0.000025 mg∙L^−1^	0.000125 mg∙L^−1^	0.00025 mg∙L^−1^
**Element**	**Standard** **Sample #4**	**Standard** **Sample #5**	**Standard** **Sample 6**	
Sc	5 g∙L^−1^	5 g∙L^−1^	5 g∙L^−1^	
Th	0.00125 mg∙L^−1^	0.0025 mg∙L^−1^	0.0125 mg∙L^−1^	
U	0.00125 mg∙L^−1^	0.0025 mg∙L^−1^	0.0125 mg∙L^−1^	

**Table 3 materials-16-03023-t003:** Standard deviation (SD) and relative standard deviation (RSD) of analysis elements.

**Element**	**Th**		
	Average concentration measured/(g∙L^−1^)	SD/(g∙L^−1^)	RSD/%
Sample #1	0.001276	0.000004	0.310
Sample #2	0.002542	0.000007	0.280
Sample #3	0.012783	0.000034	0.270
**Element**	**U**		
	Average concentration measured/(g∙L^−1^)	SD/(g∙L^−1^)	RSD/%
Sample #1	0.001253	0.000034	2.730
Sample #2	0.002514	0.000034	1.370
Sample #3	0.012631	0.000029	0.230

**Table 4 materials-16-03023-t004:** Th and U test results of spiked recovery experiment.

**Sample No.**	**Concentration** **of Th (g∙L^−1^)**	**Adding Standard Matter Amount of Th (g∙L^−1^)**	**Measured Value of Added** **Standard of Th (g∙L^−1^)**	**Recovery Rate** **of Th (%)**
1	0.00772	0.001	0.00104	103.50
2	0.00190	0.001	0.00103	103.00
3	0.00082	0.001	0.00096	95.50
4	0.00043	0.001	0.00098	98.00
5	0.00008	0.00002	0.00002	100
**Sample No.**	**Concentration** **of U (g∙L^−1^)**	**Adding Standard Matter Amount of U (g∙L^−1^)**	**Measured Value of Added** **Standard of U (g∙L^−1^)**	**Recovery Rate** **of U (%)**
1	0.01266	0.001	0.00107	107.00
2	0.00311	0.001	0.00105	105.00
3	0.00123	0.001	0.00105	105.00
4	0.00062	0.001	0.00097	96.50
5	0.00017	0.0001	0.00009	90.00

**Table 5 materials-16-03023-t005:** Standard deviation of parallel test Th, U and detection limit and quantification limit of test elements (g∙L^−1^).

Element	SD	Limit of Detection	Limit of Quantification
Th	0.0000047	0.0000142	0.0000472
U	0.0000197	0.0000592	0.0001973

**Table 6 materials-16-03023-t006:** The concentration of Th and U in seven groups of feed solution set to test the influence of scandium matrix concentration on the detection of Th and U.

Element	Sample #1	Sample #2	Sample #3	Sample #4	Sample #5	Sample #6	Sample #7
Sc	0	1	2	3	4	5	6
Th	0.0025	0.0025	0.0025	0.0025	0.0025	0.0025	0.0025
U	0.0025	0.0025	0.0025	0.0025	0.0025	0.0025	0.0025

**Table 7 materials-16-03023-t007:** Influence of scandium matrix on determination of Th and U.

Sample No.	Th (g∙L^−1^)		U (g∙L^−1^)	
	Measured Value	Relative Error	Measured Value	Relative Error
1	0.003270	30.8%	0.002695	7.8%
2	0.002860	14.4%	0.002615	4.6%
3	0.002745	9.8%	0.002575	3.0%
4	0.002630	5.2%	0.002515	0.6%
5	0.002555	2.2%	0.002490	0.4%
6	0.002465	1.4%	0.002495	0.2%
7	0.002340	6.4%	0.002380	4.8%

## Data Availability

Data will be made available on request. The interested readers can contact the corresponding authors to obtain research data.

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
