# Peer review of "Determination of Trace Thorium and Uranium Impurities in Scandium with High Matrix by ICP-OES"

_materials, 2023, doi:10.3390/ma16083023_

Round 1
Reviewer 1 Report
Abstract
Line 18: It is Inductively Coupled Plasma Optical Emission Spectrometry. There is no ionisation there. Plasma is responsible for excitation of the chemical individuals, not for an ionisation which is unwanted.
Line 20: Please correct this sentence is so chaotic. What does it mean "other techniques"?
Line 21-25: Please improve the language, I can hardly understand the idea over here.
Table 1. please correct the title and move the table to the experimental section
Lines 84-85: Please improve the language.
2.2 Reagents
Wrong calculations. 15.333g Sc2O3 in 100ml = 10g Sc in 100ml, that means that 1ml contains 0,1g Sc = 100mg, not 50mg.
line 106. It is exactly the opposite. The greater the matrix effects, the higher the detection limit.
Maybe the authors mean that the lower the dilution of the sample, the lower the detection limit? But this is a very troublesome generalization.
Line 114. The word "can" should probably be removed, because it changes the meaning of this sentence by 180 degrees?
Line 122. Please use mg/L.
The same applies to Table 2.
3.2. Precision test
Please explain why there is almost a tenfold difference in precision between the elements?
3.3 Accuracy
Recovery itself describes the trueness of the result, not the accuracy.
3.4.
The correct expressions are: Limit of detection and Limit of quantification.
The presented method of calculating the limit of quantification is incorrect and has not been used for a long time. To confirm the limit of quantification, the recovery value and the coefficient of variation at the limit of quantification level should be tested and evaluated.
Table 6 is unnecessary
Table 7: Please include results obtained without Sc addition. At the first glimpse, from the data from table 7 it looks like you method has a systematic bias, which is accidentally corrected by the increasing matrix level. How was the calibration curve made in this recovery test?
What was the concentration of Sc in the standards used for it? From your description I guess you want to show that if you equalize Sc concentration levels in the standard and the sample, you get the best recovery. But you have to clearly write what is the concentration of Sc in the standards and why did you choose 5 g/L? What it comes from? Otherwise, I may ask, what is the course of this relationship, if the standards had, for example, 3g/L?
What are these parameters for a solution without a matrix? How to explain that 1g/L causes large matrix effects and 6g/L does not?
Lines 249 and 250. What are the recovery and the RSD at the level of the Limit of quantification?
Author Response
The authors appreciate your comments and suggestions, which are very inspiring and enlightening. We have revised the article to your request. And a point-by-point response to your comments is attached as a MS Word. Please see the attachment.

Reviewer 2 Report
The present manuscript entitled "Study on the influence of the preparation method of Konjac glucomannan-silica aerogels on the microstructure, thermal in-sulation, and flame retardant properties Determination of Trace Thorium and Uranium Impurities in Scandium with High Matrix by ICP-OES " examines the to develop a technique that can accurately detect trace Th and U in high concentrations of scandium solution in the research on high-purity scandium oxide quality detection and the removal of trace impurities respectively. Here adopted some advantageous initiatives to develop a method for the inductively coupled ionization atomic emission spectrometry (ICP-OES) determination of Th and U in high-concentration scandium solutions, such as spectral line selection, matrix influence analysis, and spiked recovery. These data are interesting however, I am providing some minor comments (related to the Introduction and the results & discussion) and then a list of these comments that need to be addressed. Please find below these comments. Overall, the quality of this paper is suitable for publication in materials in terms of presentation, content, and description.
1. The introduction reads nice and adequate but the motivation part not enough. Authors should provide a clear explanation about the reason for ICP-OES with references.
2. The findings of the precision test indicate that the relative standard deviation RSD of Th < 0.4%, and the relative standard deviation RSD of U <3% in each sample for the clear understanding of the readers can mention.
3. It is recommended to why relative error exist about 30.8% in table 7 Discuss in detail.
4. It is suggested that the author can analyze it more deeply and clarify the impacts Sc concentration of 5 g∙L-1.
5. Compare the present results with using other methods determination.
Author Response

(The authors gave the same response as above.)

Round 2
Reviewer 1 Report
Thank you for considering my comments.
There are some more:
Line 67: ICP AES should be changed to ICP OES.
Line 207: The appropriate term that should be used is: Trueness.
There is no such a term in analytical chemistry like "Authenticity". Please change.
Line 213: I still believe that the LOQ is estimated incorrectly and in an unacceptable way. Such estimated limit of quantification may lead to significant errors in the interpretation of the results. I suggest not to use such calculated limit of quantification in your everyday work. It is necessary to check the recovery value at the limit of quantification level. Only then will you know the actual characteristics of the analytical signal at this concentration range (especially since you are working with matrix samples and without an internal standard).
Author Response
Thank you for your reply again. Please see the author's point-by-point respons in the attachment.
